# Virtual Behavioral Intervention to Promote Healthy Lifestyle Behaviors: A Feasibility RCT during COVID-19 Pandemic

**DOI:** 10.3390/healthcare11010091

**Published:** 2022-12-28

**Authors:** Shaima A. Alothman, Mohammed M. Alshehri, Alaa A. Almasud, Mohanad S. Aljubairi, Ibrahim Alrashed, Mohammad Abu Shaphe, Abdullah F. Alghannam

**Affiliations:** 1Lifestyle and Health Research, Health Science Research Center, Princess Nourah bint Abdulrahman University, Riyadh 11321, Saudi Arabia; 2Physical Therapy Department, Jazan University, Jazan 45142, Saudi Arabia; 3Medical Research Center, Jazan University, Jazan 45142, Saudi Arabia; 4School of Psychology, University of Queensland, St Lucia 4072, Australia

**Keywords:** behavior change, health promotion, COVID-19, lifestyle

## Abstract

Background: the COVID-19 pandemic has had a substantial impact on human health, affecting many lifestyle behaviors such as physical activity, sedentary behavior, dietary habits and sleep. Purpose: to assess the feasibility of six sessions of a virtual behavioral intervention to promote healthy lifestyle practices during a stay-at-home advisory phase of the COVID-19 pandemic. Methods: A participant-blinded randomized controlled trial was performed through a virtual platform setting. Participants were randomly assigned into two groups. They were assigned to a motivational interviewing (MI) intervention or attention group, with pre- and postintervention assessments. The MI treatment consisted of six sessions (twice each week). The same number of virtual structured sessions were provided for the attention group, and they provided brief advice to promote healthy lifestyles. The study was conducted from April to June 2020. Results: The feasibility outcomes indicated that 39 of the 50 participants (78%) completed the trial. The dropout rate was 21.7% for the attention group and 22.2% for the intervention group. Participating in MI had a significant positive interventional effect on physical activity level, distress and fear of COVID-19. Conclusions: It is feasible to deliver behavioral change interventions virtually. Further, MI can be used as a useful strategy for the favorable promotion of a healthy lifestyle. Trial registration: NCT05392218 (26/05/2022).

## 1. Introduction

SARS-CoV-2 (COVID-19) mainly affects the respiratory system and can lead to life-threating complications [1,2]. Further, there is no promising treatment for COVID-19; thus, several countries have instituted a partial or full lockdown to slow down the spread of COVID-19 [3]. In Saudi Arabia, these procedures included switching to distance learning, working from home and limiting clinical visits [4]. The restrictions that occur in regular daily life due to the implemented isolation measures may increase the difficulty of adopting a healthy lifestyle.

The impact of the COVID-19 pandemic has shown associations with mental and psychological health [5,6]. Restrictions to normal daily life before the COVID-19 era have been associated with worse psychological wellbeing [7,8,9]. Previous studies reported poorer wellbeing for people who feel isolated, have a fear of contracting a dangerous virus or practiced social distancing [10,11,12,13]. In addition, time spent in quarantine has been positively linked with more symptoms of stress, anxiety and depression [14,15,16]. Therefore, people with different health issues might face excessive worries about their health and limit their engagement in healthy lifestyle behaviors.

Healthy lifestyle behaviors include being physically active, practicing healthy eating habits, having a good quality sleep duration, reducing stress levels, engaging in positive social connections and avoiding substance abuse [17]. Engaging in one or more of the opposite healthy lifestyle behaviors (unhealthy lifestyle behaviors) has been associated with an increased risk of cardiovascular diseases and metabolic disorders incidence rate [18,19]. Further, it has been associated with higher all-cause mortality [20]. Overall, unhealthy lifestyle behaviors have contributed greatly to increasing the global health and economic burden of chronic disease incidence and complications [21].

Several therapeutic options such as health promotion or behavioral change strategies are utilized to enhance a healthy lifestyle. Interventions targeting behavioral changes utilize several theoretical models, one of which is the transtheoretical model [22,23]. In this model, the behavioral change in an individual will undergo five stages of change: precontemplation, contemplation, preparation, action and maintenance. Thus, interventions should identify which stage an individual is currently in to provide targeted strategies of behavioral therapy. Transtheoretical models have been used extensively to induce lifestyle behavioral changes in different populations with encouraging results [24,25,26]. One tool to induce behavioral change based on the transtheoretical model is motivational interviewing (MI). The MI goal is to support an individual’s effort in changing their behavior through building intrinsic motivation and clearing any indecisiveness [27]. MI has been used successfully in healthcare [28]. 

Strategies to prevent unhealthy lifestyle behaviors during lockdown are imperative to promote physical, mental and psychological health. While health organizations have dictated their time to establish vaccinations to minimize the risk of COVID-19, applying effective procedures to optimize lifestyles is warranted. Improving step counts, sleep duration and diet are important elements in lifestyle behaviors to increase immunity against viruses. However, with restricted physical communication during quarantine, several health organizations delivered health services virtually or online. This quick transition might show challenges to clients or health providers, in which future studies are needed to measure the perception of change in lifestyle behaviors using virtual health services. 

Studies exploring the use of behavioral change interventions to improve lifestyle behaviors in Saudi Arabia are scarce [29,30]. Further, health services delivered virtually or online are relatively new in practice in the region. Thus, before launching a full-scale RCT of a virtual behavioral intervention to promote healthy lifestyle behaviors, it is imperative to test the feasibility of conducting such a study. Therefore, this study aimed to investigate the feasibility of a virtual behavioral intervention to promote lifestyle behaviors during the COVID-19 pandemic.

## 2. Materials and Methods

### 2.1. Study design and Participants

This study was approved by the Princess Nourah bint Abdulrahman University Institutional Review Board (log # 20-0142). Potential participants were recruited (April to June 2020) from a cross-sectional research study (Lifestyle Behaviors during COVID-19, n = 554) that was ongoing at the time [31]. Participants who completed the main study were invited to participate in this study. Participants who expressed interest were called via phone, whereby we explained the study and gave them the opportunity to ask any question before they were directed to sign the informed consent form electronically. Participants were included in the study if they were living in Saudi Arabia with a stay-home advisory implemented. Individuals were excluded if they had a confirmed or suspected COVID-19 diagnosis to avoid potentially confounding the results. 

A participant-blinded randomized controlled trial was performed through a virtual platform setting (Figure 1). Participants were randomly assigned into two groups using a computer-generated randomization program (www.graphpad.com, accessed on 22 December 2022). Participants were assigned, by cohort, to a motivational interviewing intervention or attention group, with a pre- and postintervention assessment. Participants received no monetary incentives (clinical trials registration NCT05392218).

### 2.2. Intervention—Motivational Interviewing (MI)

A certified lifestyle medicine specialist provided training on intervention delivery using a training manual, role play and didactic instruction. All interventional sessions were based on the transtheoretical model [23] and the 5As tool (ask, assess, advice, agree and assist), which is used for brief interventions to encourage behavior change [32]. The MI treatment consisted of six sessions twice each week for up to 30 min. of individual sessions. Each session contained a different component and topic that included substance abuse, physical activity, healthy sleep, stress management, nutrition and social support. The components’ order was randomized for each participant. The interviews were conducted using a video meeting platform (Zoom Video Communications Inc., 2016) with both the video and audio function enabled, while participants were given the choice to refuse video calls should they desire. Alternative session delivery methods such as phone calls were used and documented based on participant request. 

The intervention session began with an introduction to the components of a healthy lifestyle based on The American College of Lifestyle Medicine (ACLM) model [17]. Participants’ health profile and session plan were discussed during their first session. Therapists shared their screen and showed an educational material document with the participant depending on the session topic. At the end of each session, the therapist recorded the total session time, feedback, general wellbeing, level of satisfaction and any technical problem that occurred during the interview. Upon completion of all the sessions, the participants were asked to complete a postassessment questionnaire after five days from the last session. Participants who did not respond after seven days of sending the survey were marked as nonrespondents. 

### 2.3. Attention Group

Participants in the attention control condition received the same number of virtual sessions as the MI group. However, the structure of the sessions consisted of brief advice to promote a healthy lifestyle delivered by different therapists. The therapist firmly and clearly discussed general lifestyle health topics and provided educational materials during each session. The participants were also asked to complete the postassessment questionnaire within five days of completing the last session. 

### 2.4. Assessment

#### 2.4.1. Feasibility Outcomes

Feasibility was assessed by calculating the recruitment rate, retention rates, session duration, days between sessions, technical problems and satisfaction level. The recruitment rate was reported as the number of participants who consented divided by the number of people that were invited to participate. The retention rate was reported as the number of participants completing all the study sessions. The session duration was reported as the average session time for the intervention group or attention group. The days between the sessions were calculated as the average number of days between the sessions. Technical problems were reported as the number of technical problems while conducting the session virtually. Lastly, the satisfaction level was assessed by recording the participants’ response to the following question: “are you satisfied with the session content and delivery?” 

#### 2.4.2. Anthropometric Measurements and Demographic Variables

Participants were asked to self-report their anthropometric measurements and demographic information via a standardized questionnaire due to lockdown constraints prohibiting physical measurements. Anthropometric measurements included height and body weight to calculate the participants’ body mass index (BMI). The reported demographic variables were age, smoking status, educational level and presence of chronic diseases.

#### 2.4.3. Lifestyle Behaviors and Fear of COVID-19

The outcome variables measured subjectively in this study were physical activity, sedentary behavior, psychosocial distress, sleep quality, dietary habits, social support and fear of COVID-19. Physical activity and sedentary behavior were self-reported by the participants by completing the Global Physical Activity Questionnaire (GPAQ) [33,34]. The GPAQ comprises 16 questions. The participants were classified as having low, moderate or high physical activity. Sedentary behavior was reported as the total hours spent sitting. Psychological distress was measured using the Kessler Psychological Distress Scale (K10) [35]. The scale included 10 items that concerned negative emotional states (e.g., feeling depressed, nervous or worthless) for the duration of the 4 weeks. The score ranges from 10 to 50, and higher scores reflect higher psychological distress. Sleep quality and duration were assessed using the Pittsburgh Sleep Quality Index (PSQI) [6]. The PSQI consists of 19 items that produce a global sleep quality score covering 7 components that produce one global score that ranges from 0 to 21. Poor sleepers have a score of ≥5 as a cut-off global score. Sleep duration was reported as the total hours spent sleeping. The measurement of dietary intake was conducted using a self-administrated food frequency questionnaire (FFQ) [36,37]. This retrospective assessment tool requires respondents to report specific food habits, such as the frequency of consumption of fruits and vegetables, caffeine, dairy products and the use of fats in cooking. This tool has been shown to have good validity for ranking nutrient intake and assessing dietary intake for a large population. Social support was evaluated using a validated Arabic version of the Medical Outcomes Study (MOS) Social Support Survey [38,39]. This survey measures different functional aspects of social support (tangible support, informational support, affectionate support, emotional support and positive social interaction). This tool contains 19 questions, and a higher score indicates more social support. The Fear of COVID-19 Scale (FCV-19S) was administered to measure the severity of individuals’ fear of COVID-19 [40,41]. The psychometric properties (construct validity and test–retest reliability) of the used Arabic version have been found to be satisfactory. The scale consists of seven statements reflecting emotional fear reactions towards the pandemic. Responders were asked to specify their level of agreement on each statement using a five-item Likert-type scale (1 = strongly disagree to 5 = strongly agree) with score ranging from 7 to 35. A higher score indicates a higher fear of COVID-19.

### 2.5. Statistical Analysis

The descriptive statistics for the continuous variables included means and standard deviations, with frequencies used for feasibility and categorical variables. Lifestyle behavior variables were tested for normality. Normally distributed data were analyzed using the student’s t-test, and the Mann–Whitney test was used to analyze the non-normally distributed data. Categorical data were tested using a chi-square test. A statistical evaluation was performed using GraphPad Prism (version 7.04 for Windows, GraphPad Software, La Jolla, CA, USA, www.graphpad.com, accessed on 22 December 2022). The level of significance was set at alpha = 0.05.

## 3. Results

The feasibility outcomes indicated that a total of 50 individuals were randomized from 265 invited participants, resulting in an 18.9% recruitment rate. Further, the retention rate was 78.3% in the attention group while it was 77.8% in the intervention group (Figure 2). The average session duration between the groups was significantly longer in the intervention group, while the attention group reported fewer technical problems compared to the intervention group. In terms of the satisfaction level, more than 80% of the participants in both groups were satisfied with the sessions’ quality and delivery (Table 1).

No significant difference was detected in the sociodemographic data between the attention and intervention group (Table 2). The study intervention resulted in mixed lifestyle behavior outcomes (Table 3). A higher number of participants (57%) in the intervention group engaged in high physical activity (*p* = 0.03) compared to only 17% in the attention group. Further, participants in the intervention group reported significantly lower distress scores compared to the attention group. No other significant differences were found for sedentary behavior, sleep duration and quality, social support and number of meals. However, fear of COVID-19 decreased significantly in both groups.

## 4. Discussion

In this study, people in both groups showed similar attrition rates, technical issues and satisfaction levels. Further, people who received the MI (intervention) showed higher improvements in physical activity, distress and fear of COVID-19 compared to people who received a brief healthy lifestyle promotion (attention). The overall sedentary behavior was approximately reduced by 2 hours in the intervention group compared to a half an hour reduction in the control group, although this comparison was not statically significant. To our knowledge, this is the first randomized control trial that was conducted during the COVID-19 pandemic for a Saudi population. 

Online motivational programs during the COVID-19 pandemic were recommended to control the spread of the virus and minimize contact with healthcare professionals [42]. However, there is a need to assure the feasibility of these programs for quality assurance and appropriate designs [43]. We found that people who were engaged in the MI intervention had a 22.2% dropout rate compared to a 21.7% dropout rate for people who received the attention control. Previous studies found that telehealth in physical activity education had a 34% attrition rate [44]. In addition, a meta-analysis study showed a slight increase in the attrition rate of health behavior change trials compared to control groups, which might have been because of the demands to change behaviors and blindness problems [45]. There were no significant differences in the technical issue and satisfaction level between groups. Nonetheless, there were a number of technical issues in both groups that might be a contributing factor to the attrition rate and satisfaction level. Although this study used efficient technology and adapted communication to assure the delivery of interventions, other factors such as the environment, motivation, quality assessment, utilization and implementation need to be considered for future work [46]. Lastly, we could not accurately differentiate between how many participants refused video calls and chose to receive the sessions via phone calls vs. who received phone calls due to technical issues. Thus, in any future RCT with a similar methodology, we recommend that the authors keep accurate details to explore the acceptance rate of utilizing video calls as a healthcare delivery option. 

Physical activity is an essential element for a healthy lifestyle. The COVID-19 crisis limits individuals from all populations from performing daily physical activities in their daily life, which can be overcome with flexible MI and health promotion approaches that are provided virtually [47]. Our results indicated that physical activity levels increased in participants who received the MI intervention compared to participants who only received attention. Previous research showed that when participants were given the chance to set their individualized behavioral goals, similar to our study, they were more likely to succeed in increasing their physical activity level [48]. Further, our findings were consistent with studies that targeted physical activity behavioral changes through behavioral change models via telephone/internet interventions in different geographic regions [49,50]. 

Although changing sedentary behavior was not significant between groups, there was large reduction (i.e., 105 min.) in the intervention group compared to the control group (i.e., 45 min.). This might have been due to the low sample size or the actual subjectivity effectiveness, which should be considered for future studies. Consistently, in a community-based study, there was a reduction in the sitting time (i.e., 2.9% of the baseline) for the people who received theory-based counseling sessions, which was not significant when compared to the people who received usual lifestyle education [51]. Inconsistent with our findings, a pilot study found that older adults who received 6 weeks of a telehealth program that emphasized physical activity showed no improvement in physical activity and sitting duration during the COVID-19 pandemic. Our design and methodological aspects avoided several limitations that were present in the referenced pilot study, including no control group, a low sample size, high chances of comorbidities and a lack of feasibility data. 

Promoting sleep quality and quantity is essential for a better quality of life. Sleep duration increased by approximately half an hour and the sleep quality improved by a score of 0.8 on the PSQI for the experimental group, though these changes were not significant within nor between the groups. Most of the participants were at the normal cut-off of sleep quality, which might explain the small effect size of the interventions. People in the experimental program showed an improvement trend in the PSQI, although there were only 21 subjects involved in this program. Previous studies showed the clinically meaningful difference in the PSQI scores ranging from 1.4 to 4 using several statistical approaches. Due to the nature of behavioral intervention studies, there is a need to recruit people with poor sleep quality to find a minimal importance difference. People in the attention control group had normal cutoff PSQI scores (i.e., >5), which might explain the low chance to find significant changes between the groups for this study. For future studies, increasing the sample size might overcome this limitation and improve our standing regarding the effectiveness of the current intervention. 

This study has several limitations that need to be considered in future studies. The control group received an online educational program but not the same amount of attention, which might influence the subjectivity of the outcomes in this work. Future work needs to include objective measurements of physical activity, sleep quality and sedentary behavior using accelerometers. Blinding the intervention providers and participants is challenging in clinical trials. Due to the nature of this pilot RCT, blinding the intervention providers and participants was difficult; a large fund might allow future studies to have high-quality designs. Follow-up sessions are needed to assess the continuity of lifestyle improvements or maintenance. One of the important elements of pilot studies is to measure the power of the sample size for future studies based on the outcome of interest. This pilot RCT will guarantee essential information for power calculations by controlling for attrition rate and extraneous variables based on the design of future studies.

Using virtual behavioral interventions to promote a healthy lifestyle is a promising therapeutic approach globally. Further, the high economic and health burden of chronic diseases in this region calls for the exploration of cost-effective therapeutic approaches. This study assessed the first step in achieving this goal. We found that virtual behavioral interventions are feasible in our region for the public. Future studies may investigate the effectiveness of such an approach in people at risk of developing chronic diseases or that have already been diagnosed to reduce the complications rate. 

## Figures and Tables

**Figure 1 healthcare-11-00091-f001:**
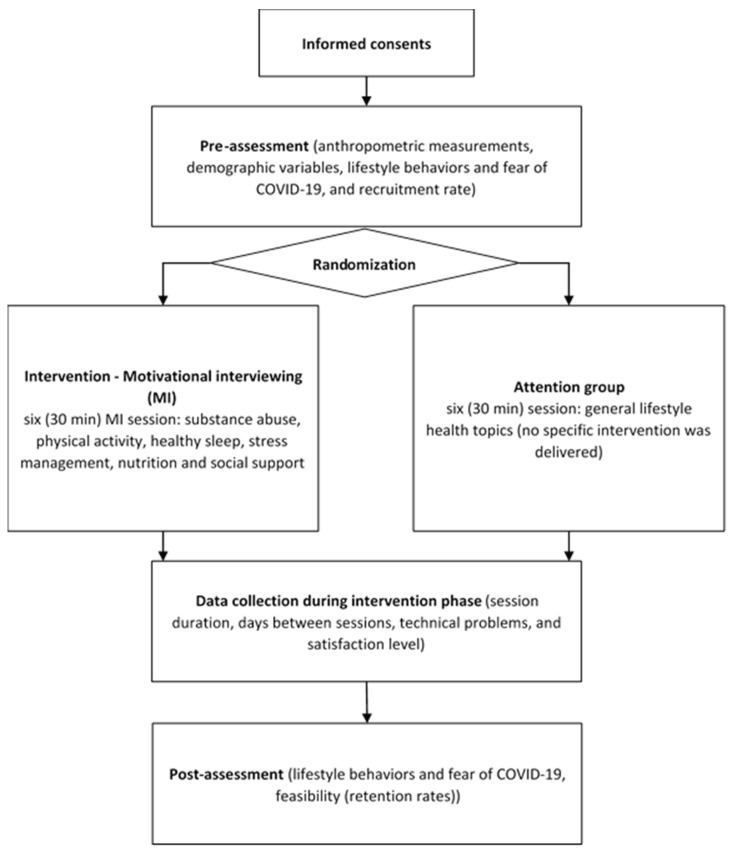
Overview of the study design.

**Figure 2 healthcare-11-00091-f002:**
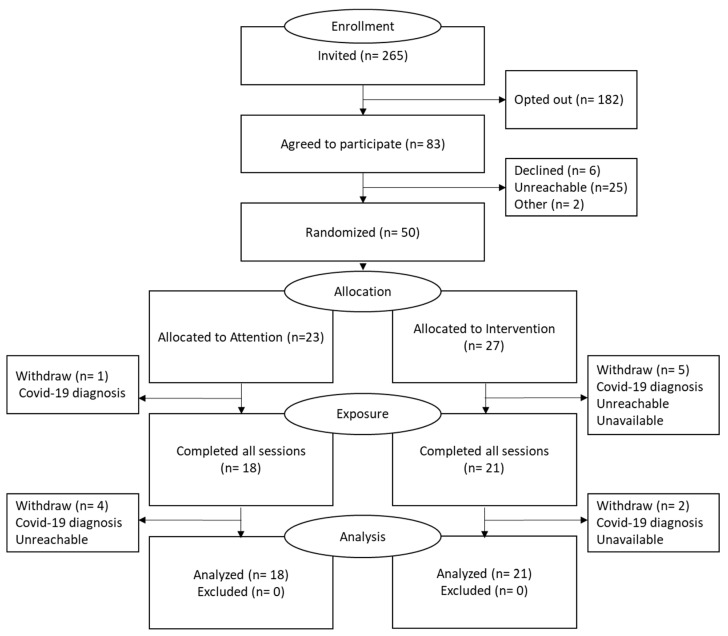
Study flow chart.

**Table 1 healthcare-11-00091-t001:** Feasibility outcomes. Data compared using student’s *t*-test or Mann–Whitney test. Data reported as mean ± SD or frequency (n).

Variables	Attention Control (n = 18)	Intervention (n = 21)	*p*-Value
Dropout Rate (%)	21.7 (5 out of 23)	22.2 (6 out of 27)	0.99
Session duration (minutes)	16.4 ± 2.6	27.4 ± 7.6	<0.01
Days between sessions (days)	2.8 ± 0.6	3.2 ± 0.8	0.10
Technical problems (count)	12.0 (13/108 sessions) 0.7 ± 1.1	19.0 (24/126 sessions) 1.1 ± 1.0	0.11
Satisfaction level (satisfied)	89.0 (16)	81.0 (17)	0.99

**Table 2 healthcare-11-00091-t002:** Sociodemographic data at baseline. Data compared using chi-square test except for age and BMI where data were compared using student’s *t*-test. Data reported as mean ± SD or frequency (n).

Variables		Attention Control (n = 18)	Intervention (n = 21)	*p*-Value
Age		37.7 ± 10.9	33.7 ± 10.2	0.24
Sex	Female %	83.3 (15)	81.0 (17)	0.8
	Male %	16.7 (3)	19.0 (4)	
BMI		26.5 ± 3.6	24.5 ± 4.8	0.16
BMI classification	Underweight	0.0 (0)	9.5 (2)	0.5
	Normal	44.4 (8)	47.6 (10)	
	Overweight	33.3 (6)	28.6 (6)	
	Obese	22.2 (4)	14.3 (3)	
Education level	Undergraduate degree or less %	72.2 (13)	71.4 (15)	0.9
	Post graduate degree %	27.8 (5)	28.6 (6)	
Number of comorbidities (i.e., diabetes, HTN, etc.)	0	72.2 (13)	52.4 (11)	0.3
	1	16.7 (3)	38.1 (8)	
	>1	11.1 (2)	9.5 (2)	

**Table 3 healthcare-11-00091-t003:** Lifestyle behavior variables and related outcomes. Data compared using student’s *t*-test, Mann–Whitney test or chi-square test. Data reported as mean ± SD or frequency (n). T1 is preassessment. T2 is postassessment. ^i^
*p*-value results of comparing attention control (T2) vs. intervention (T2).

Variables		Attention Control (n = 18)			Intervention (n = 21)			*p*-Value ^i^
		T1	T2	*p*-Value	T1	T2	*p*-Value	
Physical Activity (Level)	Low	38.9 (7)	50.0 (9)	0.78	33.3 (7)	28.6 (6)	0.11	0.03
	Moderate	50.0 (9)	33.3 (6)		42.9 (9)	13.4 (3)		
	High	11.1 (2)	16.7 (3)		23.8 (5)	57.1 (12)		
Sedentary Behavior (Hrs.)		7.6 ± 5.0	6.8 ± 3.6	0.94	7.3 ± 4.7	5.5 ± 4.3	0.15	0.20
Sleep	Duration (Hrs.)	7.7 ± 1.7	7.9 ± 2.1	0.34	7.7 ± 1.8	8.2 ± 1.6	0.22	0.76
	Quality	5.2 ± 2.6	5.5 ± 3.0	0.81	5.9 ± 2.4	5.1 ± 3.4	0.28	0.53
Distress		21.5 ± 8.5	17.6 ± 6.2	0.14	20.8 ± 8.8	15 ± 5.1	0.01	0.13
Social Support		62.4 ± 18.5	70.1 ± 17.5	0.25	70.3 ± 19.4	77.1 ± 15.7	0.23	0.15
Meals (count)	1	11.1 (2)	16.7 (3)		14.3 (3)	4.8 (1)		
	2	66.7 (12)	11.1 (2)		52.4 (11)	4.8 (1)		
	≥3	22.2 (4)	72.2 (13)		33.3 (7)	90.5 (19)		
Fear of COVID-19		18.9 ± 3.4	16.6 ± 3.1	0.01	16.4 ± 6.2	12.5 ± 5.3	0.03	0.00

## Data Availability

The datasets generated and/or analyzed during the current study are available from the corresponding author upon reasonable request.

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
