# Peer review of "Virtual Behavioral Intervention to Promote Healthy Lifestyle Behaviors: A Feasibility RCT during COVID-19 Pandemic"

_healthcare, 2022, doi:10.3390/healthcare11010091_

Round 1

Reviewer 1 Report

The topic of the article is really interesting and necessary, since it explores healthy lifestyle behaviors, but the way in which the information is presented throughout the article does not help to highlight its relevance.

The title does not reflect some relevant characteristics of the study. It would be interesting to include some reference to the COVID-19 pandemic. Moreover, it would be necessary to include the virtual nature of behavioral intervention.

Introduction section does not contextualize the topic well. First of all, the trans-theoretical model or any other base model should be explained. Increased stress and anxiety should be better related to unhealthy behaviors. In addition, it would be advisable to better explain unhealthy behaviors. It should be indicated if there is previous research in this regard and, if so, indicate the main results. Finally, it is not understood what the final goal is. Title includes the term feasibility, but no reference to it is made in this section except when introducing the objective, which is not entirely clear either.

Some comments are presented related to Materials and Methods section. Regarding the sample, Inclusion and exclusion criteria are unclear. Moreover, the reason for excluding people with confirmed or suspected COVID-19 diagnosis should be explained. Authors state that “Alternative session delivery method, such as phone calls were used and documented based on participant request”. However, no data are provided on how many people received the intervention through alternative session delivery method. Has this variable been considered in the analysis of the results? If it has been considered, it should be indicated. If it has not been considered, it is recommended to allude to it and provide some kind of reflection on it in the discussion. A comparative figure of the intervention between experimental and control group would be helpful. Is the way to assess feasibility the one that is usually used? In this case, indicate other investigations in this regard. If the evaluation format is not standardized, it is recommended to allude to it and provide some kind of reflection on it in the discussion. Instruments (2.4.3 sub-section) should be described in the order in which they are presented in the first sentence " The outcomes variables measured subjectively in this study were physical activity, sedentary behavior, psychosocial distress, quality of life, sleep quality, dietary habits, social support, and fear of COVID-19".

Results section would benefit from figures representing the main results, at least the significant ones.

Regarding to Discussion section, conclusions about the feasibility is not clear. Moreover, author state “This study had several strengths; yet, there are few limitations to be considered in future studies”, but strengths seem not to be indicated. This section should be ended in another way, indicating the main conclusion of the study, for example.

Finally, spelling errors should be checked as in the sentence “However, fear of 176 COIVD-19 decreased significantly in both groups” in 176 and 177 lines.

Reviewer 2 Report

The manuscript is mainly clear and understandable, and written on an important topic in a relatively under-discussed style. However, while reading the manuscript, some small details caught my eye.

1.       According to the reviewer, the abstract could be more scientific and significantly more precise and concrete both in terms of methodology and results. For example, the wording of the „Purpose“ remains unclear, and the sentence "Participating in MI revealed a significant interventional effect on 23 physical activity level, distress, fear of COVID-19." does not provide any clarity as to which directional changes took place.

2.       In the methodology chapter, the sentence "Informed consents were obtained electronically 59 from each participant prior to the study" raises a question. How was the consent obtained anyway? Did the subjects give INFORMED consent, or did they receive answers to their questions in advance if necessary?

3.       How was the "satisfaction level" of the subjects evaluated? (line 111)

4.       Were the subjects instructed when measuring height? (line 115)

5.       In what form was the presence of chronic diseases investigated (interview, standardized questionnaire)? (line 117)

6.       When presenting the results, it remains unclear what the P-value (the very last column) shows anyway?

7.       In line 177, instead of COIVD-19, write COVID-19!

8.       Line 185 instead of People write people!

9.       The discussion remains shallow at times.

1-   Another limitation is the fact that no difference between the sexes was studied, so based on the literature there are definitely differences in the level of physical activity as well as in distress and social support. Or do you have data for Saudi Arabia that there is no gender difference?

1-   There is no conclusion at the end of the manuscript.

I wish all the best!

Round 2

Reviewer 1 Report

Thank you for attending to most of my suggestions. I consider that the article has clearly improved and is ready for publication. Regarding the figures for the presentation of the results, I consider that they are always necessary, since visually they bring the reader closer in a very direct way.

All the best.